# Synergistic Effect of Combined Treatment with Allicin and Antioxidant of Bamboo Leaves and Preservation of Bullfrogs (*Lithobates catesbeiana*) during Refrigeration Storage

**DOI:** 10.3390/foods12183467

**Published:** 2023-09-18

**Authors:** Weiqing Lan, Bingjie Zhang, Jintao Du, Shengyun Zhu, Xiao Xu, Jing Xie

**Affiliations:** 1College of Food Science and Technology, Shanghai Ocean University, Shanghai 201306, China; kid14121117@163.com (B.Z.); djt_2020@163.com (J.D.); wonder_hei@163.com (S.Z.); 2Shanghai Aquatic Products Processing and Storage Engineering Technology Research Center, Shanghai 201306, China; 3National Experimental Teaching Demonstration Center for Food Science and Engineering, Shanghai Ocean University, Shanghai 201306, China; 4Jiangsu Zhongyang Group Limited by Share Ltd., Nantong 226600, China; xuxiao@zyfwf.cn

**Keywords:** *Lithobates catesbeiana*, allicin, antioxidant of bamboo leaves, quality, shelf life, refrigerated storage, antimicrobial activity

## Abstract

The effects of allicin and antioxidant of bamboo leaves (AOB) on the quality of bullfrogs (*Lithobates catesbeiana*) during refrigerated storage (4 °C) were investigated. The quality changes in samples treated with deionized water (CK), allicin solution (All), antioxidant of bamboo leaves (AOB), and allicin solution combined with AOB solution (AA) in microbiological, physicochemical, and sensory evaluation were analyzed, respectively. The results demonstrated that combination treatment inhibited the increase in total viable counts, delayed the decrease in amino acid content, and retarded the sensory deterioration. Preservative treatment has an inhibitory effect on the early storage of PBC, which can reduce PBC by about 1.0 log CFU/g. The reduction in thiobarbituric acid (TBA) content and total volatile basic nitrogen (TVB-N) content indicated that combination treatment could better restrain the lipid oxidation and degradation of protein than the CK group and single-treatment group. In addition, the TVB-N content in the AA group still did not exceed the threshold on the 14th day. As a consequence, combination treatment prolonged the shelf life of bullfrogs for another six days. Therefore, allicin and AOB with excellent antioxidant and antimicrobial activity could be an effective approach to delay the biochemical reaction of refrigerated bullfrogs. This study has provided a potential approach for increasing the shelf life of bullfrogs and preserving their quality during refrigerated storage.

## 1. Introduction

Bullfrogs (*Lithobates catesbeiana)* are a type of aquacultural species in China and are considered as one of the most expensive cultured frog species. Cooked bullfrog meat is white in color, tender in texture, and lightly sweet in taste, and is similar in mouthfeel to chicken [1]. Bullfrog legs are popular among consumers due to their high nutritional value. Specifically, they are rich in moisture and protein and low in calories. They are considered to be delicious food and are popular in many countries around the world [2]. Bullfrog meat is popular not only because of its delicious taste but also because it is rich in high-quality proteins that are easily digested and abundant in many essential amino acids [3]. Compared with frozen meat (especially after frozen–thawed cycles), fresh bullfrog meat is preferred by customers. Compared with the fineness and juiciness of fresh bullfrog meat, the frozen–thawed meat has diminished sensorial qualities of taste after cooking. Therefore, local breeders usually take live bullfrogs to markets rather than sacrificing them. However, live-bullfrog transportation is also flawed because the bullfrogs can be easily frightened, causing them to fatally trample each other. Moreover, cannibalism is more frequent in crowded space during transportation and subsequent processing [4]. As a result, the development of post-slaughter processing based on refrigeration is urgent for the bullfrog industry.

Allicin (All) is an oxygenated sulfur compound, which was isolated from crushed garlic petals in 1944 and identified as a compound with antibacterial activity. As one of the important organic sulfur compounds, it is beneficial to human health but is also accompanied by a typical penetrating odor [5,6]. The interaction of alliin (S-allyl-L-cysteine sulfoxide) with the catalytic action of alliinase (alliin lyase; EC 4.4.1.4) during the crushing of garlic leads to the production of allicin.

Antioxidant of bamboo leaves (AOB) extracted from bamboo leaves, which is a brown–yellow powder, is mainly composed of lactones, flavonoids, and phenolic acids [7]. It is a natural additive and has been widely used in food industry [8]. AOB is widely used in food preservation, such as in the preservation of fish, meat, and edible oils. Some research has shown that AOB has good antioxidant activity on the preservation of aquatic products [8,9,10].

At present, fresh delivery is very popular in China. As one kind of fresh food, bullfrog quality is particularly critical in the process of cold-chain delivery. Frozen bullfrog products undergo severe moisture loss and possess poor taste. The cost of live-bullfrog transportation is expensive. Therefore, it is important to develop a post-slaughter bullfrog storage method with refrigeration as the mainstay. However, most of them are thermal treatments; there are no studies on the effects of natural biopreservative treatments on bullfrogs. The objective of our study is to investigate the effects of All and AOB treatment during refrigerated storage on the quality of bullfrogs.

## 2. Materials and Methods

### 2.1. Materials and Reagents

The allicin (purity: ≥95%) was purchased from Aladdin Industrial Inc., Shanghai, China. AOB (≥40.0% of total phenols) was provided by Shaanxi Zhenghe Phar Biotechnology Co., Ltd. (Xi’an, Shaanxi, China). All reagents used in the study were analytical grade.

### 2.2. Preparation of Bullfrog Samples and Sample Treatment

Live bullfrogs were purchased from an aquatic market (Shanghai, China) and transported to the laboratory immediately. Subsequently, samples were sacrificed. The head and internal organs were removed, and then the sample was flushed with water. They were stochastically split into four groups and underwent different treatments: (1) samples treated with deionized water (CK); (2) samples treated with 0.5 g/L (*w*/*v*) of allicin solution (soluble in acetic acid solution) (All); (3) samples treated with 0.5 g/L (*w*/*v*) of antioxidant of bamboo leaves (soluble in acetic acid solution) (AOB); and (4) samples treated with 0.25 g/L of allicin solution combined with 0.25 g/L of AOB solution (AA). Each group was dipped in the respective solution for 10 min, drained, and then stored in polyethylene bags at 4 °C in a refrigerator for subsequent analysis at different intervals. Microbiological, physicochemical, sensory evaluation, and protein indexes were analyzed at 2-day intervals in triplicate. It should be noted that except for TBA content, which was detected using abdominal meat, all other indicators were assessed using leg muscles.

### 2.3. Microbiological Analysis

The microbials were carried using the method from Liu et al. [11]. The total viable count (TVC) and psychrophilic bacteria count (PBC) were performed on samples incubated at 30 °C for 72 h and at 4 °C for 10 days using plate counting agar (PCA). All counts were conducted in triplicate and are expressed as log CFU/g of samples.

### 2.4. Physicochemical Analysis

Two grams of bullfrog sample was homogenized with 18 mL of distilled water. The final solution was measured using a digital pH meter (Mettler Toledo, Zurich, Switzerland).

The determination of the total volatile base nitrogen (TVB-N) followed the method from Li et al. [12] with slight modification. An amount of 2.0 g of minced sample was analyzed using a FOSS Kjeldahl analyzer (FOSS China Shanghai Co. Ltd., Shanghai, China) after adding MgO. The TVB-N content is denoted by mg N/100 g. 

The TBA measurement followed by the method described by Sanchez et al. [13]. In short, 5 g of bullfrog meat was homogenized with 25 mL of trichloroacetic acid (7.5%, *w*/*v*). The mixture was centrifuged (8000× *g*) for 10 min. and the supernatant was taken. An equal amount of thiobarbituric acid solution was reacted with the supernatant and the absorbance (532 nm) was measured using a spectrophotometer. TBA contents are expressed as (MDA)/kg.

The water-holding capacity (WHC) of bullfrog samples was detected using the method presented by Zhao et al. [14]. The WHC value is expressed in Equation (1):(1)$$WHC\left(\%\right)=\frac{m_{2}}{m_{1}}\times 100$$
where m_1_ and m_2_ represent the sample weights before and after centrifugation, respectively.

TPA was measured using a texture analyzer (Stable Micro Systems Ltd., Godalming, Surrey, UK), and the parameters followed the method described in [15]. The measurement parameters were set as follows: the force arm was 30 kg, the probe descent speed before testing was 1 mm/s, the testing speed was 5 mm/s, the probe return speed after testing was 1 mm/s, the testing interval was 5 s, the compression ratio was 35%, and the triggering force was 5 g. Each group of samples was tested 6 times.

Free amino acids (FAAs) were assessed as described by Li et al. [16]. A two-gram bullfrog sample was mixed with 15 mL of cold trichloroacetic acid (5%). The mixture was centrifuged after 5 min of sonication. It was then kept static at 4 °C for 2 h, homogenized, and then centrifuged at 4 °C at 10,000× *g* for 10 min. The supernatants were combined, the pH adjusted to 2.0, and then diluted to 10 mL. The solution was filtered with 0.22 μm water phase filter then analyzed using an amino acid analyzer (model L-8800; Hitachi, Tokyo, Japan). 

### 2.5. Protein Characteristic

#### 2.5.1. Fluorescence Spectroscopy Analysis

The intrinsic fluorescence emission spectra of a 1 mg/mL protein solution were determined using an F-7100 fluorescence spectrometer (Shanghai Smaio Analytical Instruments Co. Ltd., Shanghai, China).

#### 2.5.2. Myofibril Fragmentation Index (MFI)

MFI was measured as described by Zhao [14]. In short, the myofibril solution was diluted to 0.5 mg/mL and the MFI value was indicated by the absorbance value at 540 nm multiplied by 200. 

### 2.6. Sensory Evaluation

Sensory characteristics of bullfrogs with different treatments were assessed by 10 professional trained judges (five male and five female testers from different regions) in the laboratory. The selected sensory evaluators were trained to enhance their ability to perceive, recognize, and describe sensory stimuli, enabling them to provide accurate, consistent, and reproducible sensory measurement values. Sensory evaluation was assessed as described by Lan et al. [17]. Sensory analysis was performed on each group according to color, texture, and odor, and overall acceptability scores of 40, 30, and 30 percent were weighted based on a 10-point descriptive analysis. Scores of 8~10 indicated “very good”, 5.0~7.9 corresponded to “good”, and 1.0~4.9 indicated spoiled.

### 2.7. Statistical Analysis

The results are denoted as the means ± standard deviation, and all measurements were repeated at least in replicate. The data were analyzed using SPSS software (vision 13.0) and Origin 2018. Statistical analysis was performed with Duncan’s multiple-range tests and one-way ANOVA, and significant differences were set at *p* < 0.05.

## 3. Results and Discussion

### 3.1. Microbiological Analysis

From Figure 1, it can be seen that the number of bacteria in the treated groups was lower than that in CK group (*p* < 0.05). The initial value of TVC in the CK group was 4.03 log CFU/g and reached 9.02 log CFU/g on day 12 (Figure 1a). The bacterial growth rate of the CK group was higher than those of the other groups. The above results demonstrated that All and AOB treatment could effectively inhibit the increase in microorganisms. Psychrophiles are the principal microorganisms contributing to the deterioration of fish products during storage. The initial PBC was 4.0 log CFU/g in the CK group (Figure 1b). The PBCs in the other treated groups were 2.95, 3.10, and 2.94 log CFU/g at day 0, which were significantly lower than that in the CK group, indicating that the preservative treatment could inhibit the PBC during early storage. The number of *psychrophilic* bacteria was consistent with the trend of TVC.

### 3.2. Physicochemical Analysis

#### 3.2.1. Change in pH Value and TVB-N and TBA Contents

The accumulation of alkaline substances is mainly due to the activity of endogenous enzymes and microorganisms, which causes a rise in the pH value. Volatile bases are converted from proteins and other nitrogenous substances [11]. In Figure 2a, the initial pH value of bullfrogs was 6.44, which was a little bit lower than the results reported by [18]. The increase in pH value in the CK group had the largest range of change. The pH value in the CK group varied from 6.44 to 7.27 throughout storage, whereas samples in the All, AOB, and AA groups had relatively lower pH values. In the early stage of storage, the anaerobic fermentation of glycogen in the bullfrog meat produced lactic acid, while organic acids were also produced during the citric acid cycle, leading to a decrease in pH value. In the later stage, fish protein decomposed into alkaline amino acids under the action of enzymes, and into alkaline substances such as ammonia and trimethylamine under the action of microorganisms, resulting in pH fluctuations and changes [19,20].

TVB-N is extensively used to assess the quality of aquatic products. The TVB-N content can also reflect the quality of bullfrogs, as it is closely related to the activities of bacteria and relative enzymes. The TVB-N content in the CK group increased significantly, ranging from 6.65 to 43.81 mgN/100 g (*p* < 0.05) (Figure 2b). Lang [21] reported that the acceptability limit of TVB-N is 30 mgN/100 g. However, Sohrab [22] documented that different seafoods had different levels of acceptability. In the present study, the TVB-N content in the CK group was 25.26 mgN/100 g on the eighth day. The TVB-N contents in the other groups exceeded 25.26 mgN/100 g on the 12th, 12th, and 14th days. In addition, the TVB-N content in the AA group still did not exceed the threshold on the 14th day. This indicates that the combined treatment could better inhibit the growth of microorganisms and the oxidative degradation of proteins.

TBA content is a vital criterion to evaluate lipid oxidation. Malondialdehyde (MDA) is a type of secondary lipid oxidation product. In the initial and intermediate stages, the level of the TBA active substance increased. Some substances, such as amino acids, proteins, nucleosides, and nucleic acids in bullfrogs, led to increases in the TBA content at the end of storage [16]. As illustrated in Figure 2c, the TBA contents in All, AOB, and AA groups rose from 0.06 mg MDA/kg at day 0 to 0.10, 0.09, and 0.09 mg MDA/kg at day 14, respectively. However, the TBA content in the CK group reached 0.10 mg MDA/kg on day 10. The AOB and AA treatments were valid in retarding the lipid oxidation, which may be related to the antioxidant effect of AOB (*p* < 0.05).

#### 3.2.2. Change in WHC and TPA

The WHCs of the samples are shown in Figure 2d. There was a downward trend among groups, while the CK group exhibited lower WHC compared to the All, AOB, and AA groups. The WHC in the CK group decreased from 63% to 49%, and those in the All, AOB, and AA groups decreased to 52%, 54%, and 54%, respectively. The results showed that the treatment reduced the water loss. The sharp decrease in WHC in the CK group may be explained by the degeneration of actin and myosin, which impaired the water-holding capacity [23]. The WHC in the AA group was the highest, suggesting that the combined allicin and AOB treatment could effectively reduce the moisture loss. The results showed that the treatment could protect muscle protein, reduce water loss, and maintain the quality of the bullfrogs.

To a certain extent, the acceptability of products for consumers depends on texture, because the texture properties reflect the gel properties and taste characteristics of bullfrogs. Musculature is influenced by WHC and some inherent biological factors [24]. The hardness and springiness of bullfrogs with different treatments are shown in Figure 2e,f. It can be observed that the hardness of the sample showed a downward trend. Moreover, the AA group had a higher hardness value than the CK, All, or AOB groups. After the action of an external force, the ability of the sample to be restored to its initial condition is called springiness (also named elasticity). Springiness reflects the degree of recovery of flesh after action from an external force [11]. It was found that only the CK group showed a significant decrease in springiness during the first two days of storage. After that, the springiness gradually decreased. The springiness in the AOB and AA groups decreased significantly on day six. It is noteworthy that the springiness of bullfrogs was higher after treatment with the compound solution than after those of the other groups. The results of TPA were compatible with the variation trend in WHC. The texture-protective effect of the preservative could be ascribed to its ability to inhibit the proteolytic enzyme activity in muscle tissue, which can destroy cells and cause rapid structural changes [25]. The results demonstrated that the combination treatment had a protective effect on the texture of bullfrogs during refrigerated storage.

#### 3.2.3. Free Amino Acids

FAAs can be the precursors of volatile organic compounds for aquatic products, and they are closely related to flavor. In Table 1, the sixteen detected FAAs are presented. Among all kinds of FAAs, the contents of alanine (Ala), glutamic acid (Glu), and glycine (Gly) were the top three amino acids. From day 0 to day 12, the total FAA content increased in each group, although the increments varied in size. Then, the final FAA content values reached 86.03 mg/100 g in the CK group, 81.38 mg/100 g in the All group, 72.56 mg/100 g in the AOB group, and 53.87 mg/100 g in the AA group. 

The TAV value was calculated based on different FAA thresholds [26] and reflected the effect of different treatments on the flavor property of bullfrogs (Table 2). In Figure 3, the value of umami-TAV decreased during storage. Nevertheless, at day 12, the free amino acid content increased in each treated group, which enhanced the flavor value of the bullfrogs. Bitterness-TAV showed an upward trend among groups. AA treatment could retard the increase in histidine content by controlling the microbial metabolism and water loss. Histidine content ranged from 8.38 mg/100 g to 12.17 mg/100 g in the CK group, but the corresponding contents in AA group were only 5.89 mg/100 g at day 12, which led to bitterness reduction in bullfrog meat throughout the storage period.

### 3.3. Protein Characteristics

#### 3.3.1. Fluorescence Spectroscopy Analysis

Figure 4 shows the variation in the intrinsic fluorescence intensity (IFI) of bullfrogs. The highest fluorescence intensity was observed in a fresh sample at 335 nm, signifying that the protein structure was complete because residues were enclosed inside the MP. The IFI in the myofibrillar protein (MP) reflected the protein conformational changes [14]. It can be seen that the IFI in the CK group dropped sharply throughout the storage period, which was mainly caused by the unfolding of an oxidized MP. It can be seen that the AA group had a higher IFI compared with the other three groups at the later stage of storage, which showed that the combination treatment could delay the denaturation, aggregation, or destruction of the tertiary structure in MP.

#### 3.3.2. Myofibril Fragmentation Index (MFI)

The MFI value reflects the tenderness of meat and represents the myofibril integrity [27]. A high MFI value indicates a severe disruption in the myofibril structure, which is associated with improvement in tenderness. In Figure 4e, the MFI value significantly increased from 9.90 to 32.65 in the CK group (*p* < 0.05). On day 0, the MFI values in the treated group were lower than that in the CK group, which indicated that the addition of preservatives inhibited the increase in MFI. The All and AA group showed a better effect than the CK and AOB group during storage. This may be related to the bacteriostasis of allicin, which inhibited the growth of bacteria and restrained the protein degradation. Therefore, the All and AA group had a lower MFI value, and the MFI results were consistent with the change in the microbiological index.

### 3.4. Sensory Evaluation

Sensory evaluation uses sensory characteristics as a visible index of freshness, containing texture, odor, color, and overall acceptability [28]. The overall acceptability for human consumption is considered inedible when the score is below five. The original sensory scores of all parameters were close to nine, indicating that the bullfrog meat was very fresh. Originally, as the storage time progressed, the sensory scores in Figure 5 showed a downward tendency. However, the sensory scores in the All and AA group were higher than those of others. The All and AOB samples were acceptable until the 12th and 14th days, while the overall acceptability of the AA sample remained above five on the 14th day. In contrast, the samples of the CK group were acceptable until the eighth day of storage. According to the results, sensory analysis was considered as a good indicator because its values significantly correlate with TVB-N and other indexes. In addition, the results illustrated that the combination solution treatment could maintain better quality and prolong the shelf life of bullfrog for another six days.

### 3.5. Correlation Analysis

Correlation analysis between quality indexes of bullfrogs during refrigerated storage is exhibited in Figure 6. There was a positive correlation among TVB-N, TBA, TVC, PBC, and MFI. These indicators showed negative correlations with WHC, hardness, springiness, and sensory analysis. Not surprisingly, with the extension of storage time, the increase in TVC, PBC, TVB-N, and TBA contents reflected the degradation of protein and lipid oxidation, resulting in a reduction in water-holding capacity, texture deterioration, and the decline in sensory analysis. There were significant correlations between sensory analysis and TBA, TVB-N, TVC, PBC, WHC, hardness, springiness. Allicin and antioxidant of bamboo leaves possess antioxidant and antibacterial functions, which can inhibit microbial growth and kill bacteria, delay protein and lipid oxidation, slow down the decline in the sensory quality of bullfrogs, and extend their shelf life.

### 3.6. Diagram of Antibacterial Mechanism of Allicin and AOB

Allicin can rapidly pass through cell membranes with ease owing to its hydrophobic nature, reaching cellular compartments where it reacts rapidly with free thiol groups. Acetyl-CoA synthetase blockage and glutathione oxidation initiation, which result in a shift in cellular redox potential, disrupt the membrane, leading to cellular-content leakages [5]. AOB plays a major role in scavenging free radicals (Figure 7). Based on these variations, microbial growth is effectively inhibited. Therefore, allicin and AOB could retard the increase in TVC, TVB-N, and TBA and extend the shelf life of bullfrogs.

## 4. Conclusions

The results of the present study suggested that allicin, antioxidant of bamboo leaves, and combination treatments could retard the growth of microorganisms, the augmentation of TBA and TVB-N, and the change in texture in bullfrogs during refrigerated storage. The combined treatment was effective in sensory evaluation of bullfrogs, and the sensory quality was well maintained. At the same time, the shelf life of bullfrogs during refrigerated storage could be prolonged for another six days. Therefore, the excellent antioxidant and antimicrobial activity of allicin and AOB, as revealed in the present study, provide a potential approach for increasing the shelf life of bullfrogs and preserving their quality during storage. However, the mechanism by which allicin and AOB improve quality and inhibit the specific spoilage microorganisms in bullfrogs needs to be further investigated.

## Figures and Tables

**Figure 1 foods-12-03467-f001:**
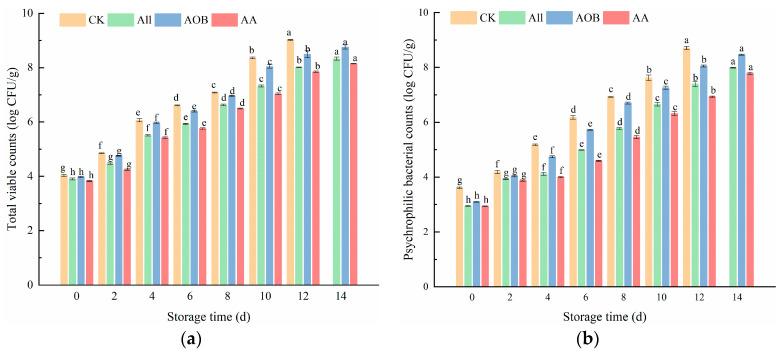
Changes in total viable counts (**a**), *Psychrophilic bacteria counts* (**b**) of bullfrogs (*Lithobates catesbeiana*) with different treatments during refrigerated storage. Different superscript lowercase letters represent significant differences within groups (*p* < 0.05).

**Figure 2 foods-12-03467-f002:**
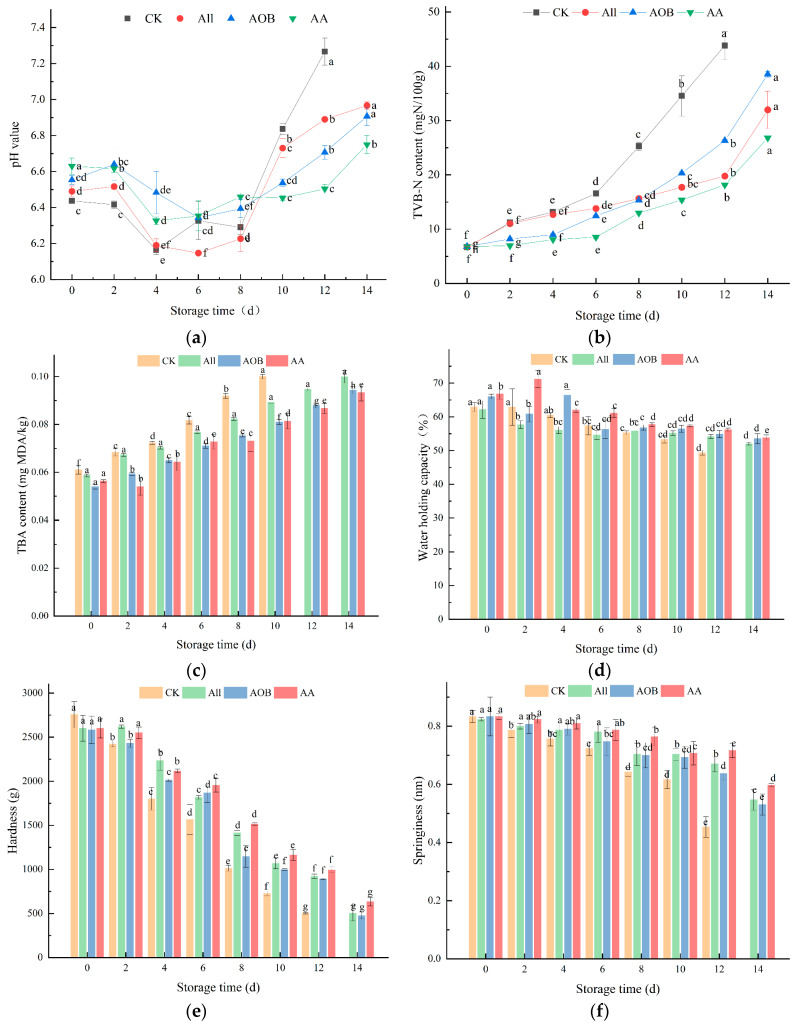
Changes in pH value (**a**), TVB-N content (**b**), TBA content (**c**), WHC (**d**), hardness (**e**), and springiness (**f**) of bullfrogs *(Lithobates catesbeiana)* with different treatments during refrigerated storage. Different superscript lowercase letters represent significant differences within groups (*p* < 0.05).

**Figure 3 foods-12-03467-f003:**
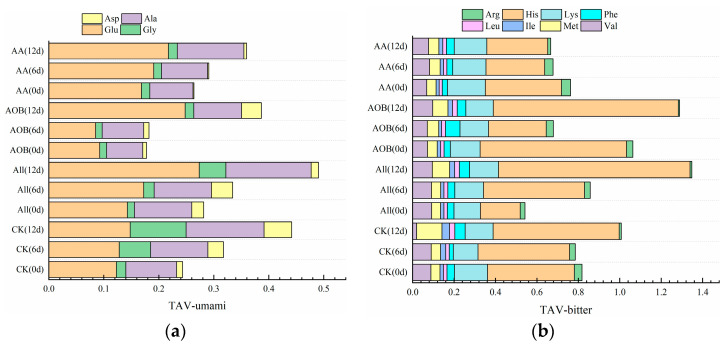
The changes in TAV-umami (**a**) and TAV-bitter (**b**) in bullfrogs (*Lithobates catesbeiana*) with different treatments during refrigerated storage.

**Figure 4 foods-12-03467-f004:**
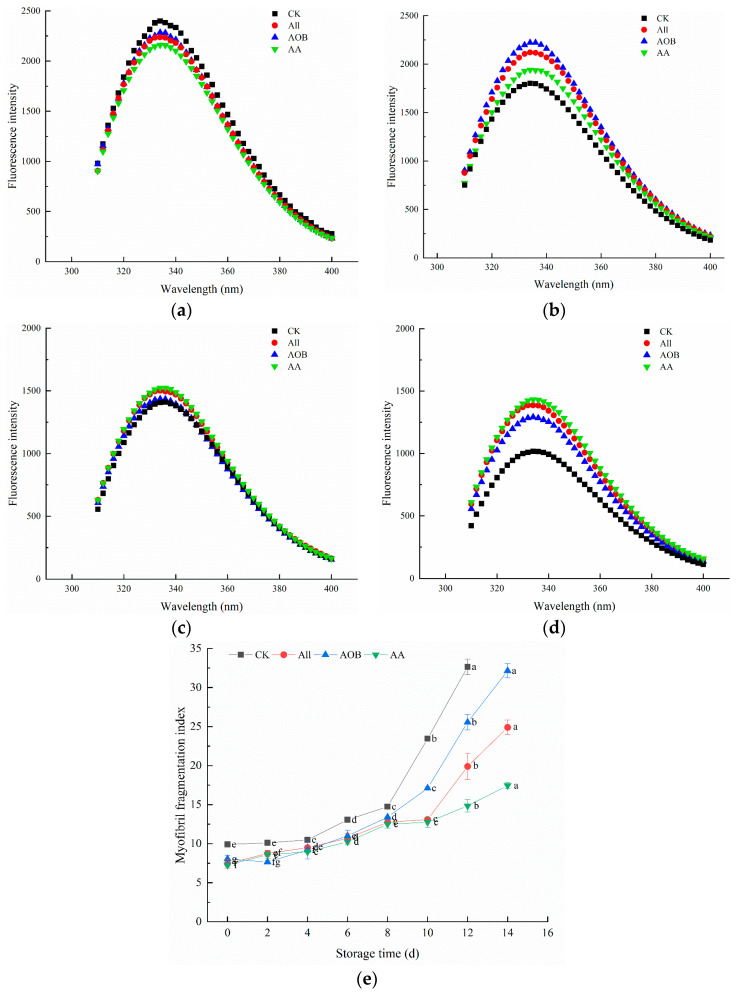
Changes in the protein tertiary structure (0 d (**a**), 4 d (**b**), 8 d (**c**), and 12 d (**d**)) and MFI (**e**) of bullfrogs (*Lithobates catesbeiana*) with different treatments during refrigerated storage. Different superscript lowercase letters represent significant differences within groups (*p* < 0.05).

**Figure 5 foods-12-03467-f005:**
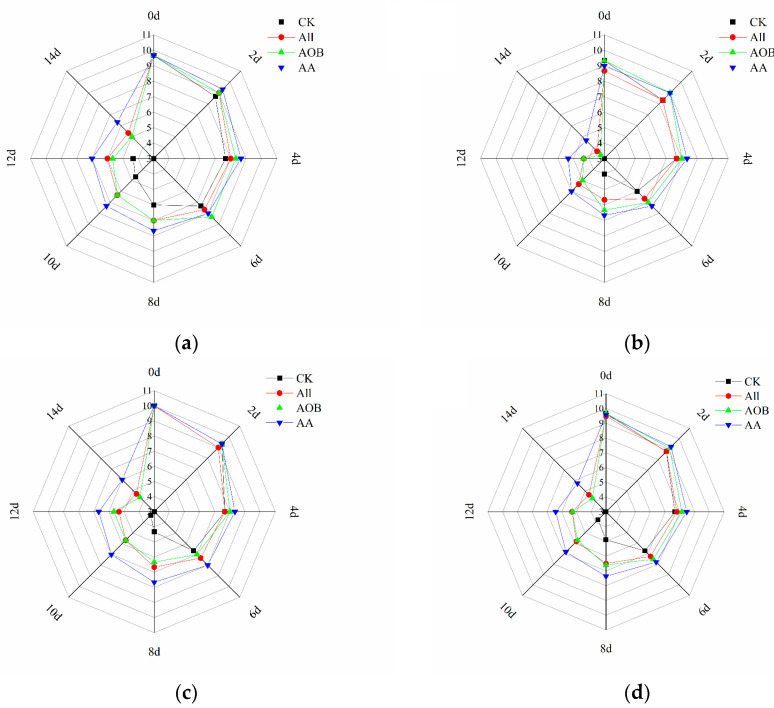
Changes in the color (**a**), odor (**b**), texture (**c**), and overall acceptability (**d**) of bullfrogs (*Lithobates catesbeiana*) with different treatments during refrigerated storage.

**Figure 6 foods-12-03467-f006:**
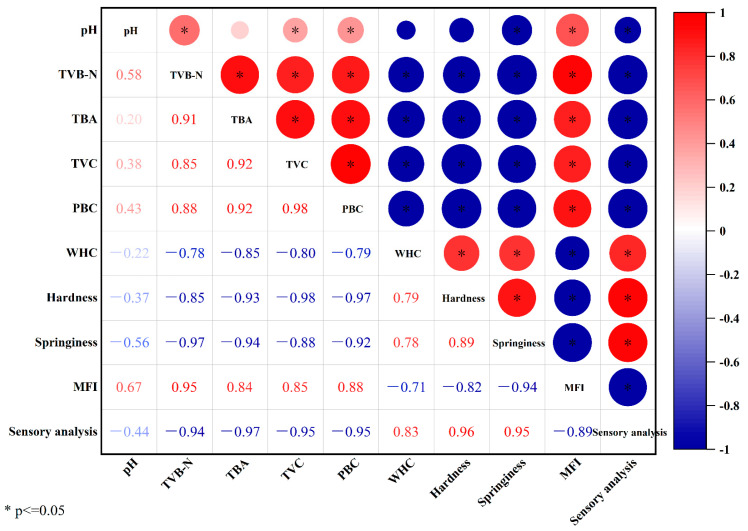
Correlation analysis between quality indexes in bullfrogs *(Lithobates catesbeiana)* during refrigerated storage.

**Figure 7 foods-12-03467-f007:**
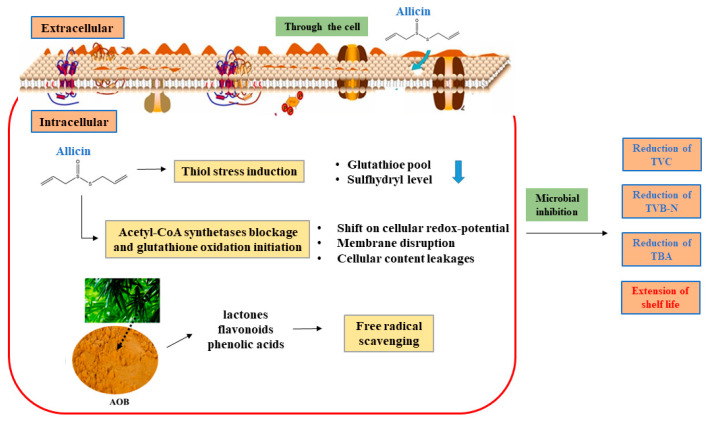
Diagram of antibacterial mechanism of allicin and AOB.

**Table 1 foods-12-03467-t001:** Changes in FAA of bullfrogs *(Lithobates catesbeiana)* with different treatments during refrigerated storage.

**Samples**	**Asp**	**Thr**	**Ser**	**Glu**	**Gly**	**Ala**	**Val**	**Met**	**Ile**
CK-0	1.09 ± 0.16 ^c^	3.08 ± 0.05 ^a^	2.63 ± 0.00 ^b^	3.68 ± 0.03 ^b^	2.24 ± 0.01 ^c^	5.54 ± 0.04 ^c^	3.54 ± 0.40 ^a^	1.31 ± 0.39 ^b^	1.47 ± 0.02 ^c^
CK-6	2.82 ± 0.02 ^b^	1.78 ± 0.02 ^b^	6.10 ± 0.03 ^a^	3.84 ± 0.03 ^b^	7.39 ± 0.05 ^b^	6.25 ± 0.00 ^b^	3.59 ± 0.00 ^a^	1.35 ± 0.02 ^b^	2.26 ± 0.09 ^b^
CK-12	5.02 ± 0.38 ^a^	1.29 ± 0.06 ^c^	6.39 ± 0.36 ^a^	4.43 ± 0.08 ^a^	13.23 ± 0.39 ^a^	8.50 ± 0.13 ^a^	0.79 ± 0.02 ^b^	3.65 ± 0.16 ^a^	3.27 ± 0.20 ^a^
All-0	2.17 ± 0.03 ^c^	2.04 ± 0.05 ^b^	1.51 ± 0.03 ^c^	4.28 ± 0.15 ^c^	1.64 ± 0.01 ^c^	6.27 ± 0.03 ^b^	3.67 ± 0.57 ^a^	1.30 ± 0.06 ^b^	1.44 ± 0.11 ^b^
All-6	3.86 ± 0.13 ^a^	1.89 ± 0.04 ^b^	3.16 ± 0.08 ^a^	5.17 ± 0.23 ^b^	2.47 ± 0.02 ^b^	6.25 ± 0.00 ^b^	3.63 ± 0.22 ^a^	1.34 ± 0.01 ^b^	1.39 ± 0.35 ^b^
All-12	1.33 ± 0.02 ^b^	2.46 ± 0.07 ^a^	1.74 ± 0.05 ^b^	8.20 ± 0.21 ^a^	6.30 ± 0.04 ^a^	9.32 ± 0.05 ^a^	3.80 ± 0.03 ^a^	2.51 ± 0.01 ^a^	2.20 ± 0.08 ^a^
AOB-0	0.67 ± 0.02 ^b^	0.67 ± 0.02 ^c^	1.76 ± 0.09 ^b^	2.78 ± 0.33 ^b^	1.62 ± 0.01 ^b^	3.95 ± 0.05 ^c^	2.86 ± 0.02 ^b^	1.41 ± 0.02 ^b^	1.49 ± 0.01 ^b^
AOB-6	0.96 ± 0.08 ^b^	1.45 ± 0.02 ^b^	1.74 ± 0.01 ^b^	2.55 ± 0.12 ^b^	1.53 ± 0.05 ^b^	4.54 ± 0.23 ^b^	2.83 ± 0.05 ^b^	1.66 ± 0.03 ^b^	1.22 ± 0.12 ^b^
AOB-12	3.60 ± 0.81 ^a^	2.19 ± 0.06 ^a^	3.50 ± 0.17 ^a^	7.43 ± 0.12 ^a^	2.07 ± 0.03 ^a^	5.21 ± 0.20 ^a^	3.87 ± 0.36 ^a^	2.20 ± 0.17 ^a^	1.99 ± 0.10 ^a^
AA-0	0.17 ± 0.01 ^c^	0.17 ± 0.01 ^c^	1.62 ± 0.24 ^a^	5.05 ± 0.83 ^a^	1.97 ± 0.27 ^a^	4.71 ± 0.34 ^b^	2.72 ± 0.04 ^a^	1.37 ± 0.37 ^a^	1.31 ± 0.03 ^c^
AA-6	0.22 ± 0.01 ^b^	2.36 ± 0.07 ^a^	1.61 ± 0.06 ^a^	5.72 ± 0.23 ^a^	1.84 ± 0.00 ^a^	5.03 ± 0.01 ^b^	3.25 ± 0.12 ^a^	1.54 ± 0.16 ^a^	1.51 ± 0.00 ^b^
AA-12	0.50 ± 0.00 ^a^	1.98 ± 0.00 ^b^	1.25 ± 0.02 ^a^	6.52 ± 0.02 ^a^	2.11 ± 0.01 ^a^	7.27 ± 0.04 ^a^	3.07 ± 0.46 ^a^	1.50 ± 0.59 ^a^	1.67 ± 0.01 ^a^
**Samples**	**Leu**	**Tyr**	**Phe**	**Lys**	**His**	**Arg**	**Pro**	**Total**	
CK-0	3.35 ± 0.09 ^b^	2.18 ± 0.02 ^c^	3.20 ± 0.09 ^ab^	7.99 ± 0.00 ^a^	8.38 ± 0.09 ^b^	1.86 ± 0.15 ^c^	2.47 ± 0.31 ^a^	54.01 ± 1.87	
CK-6	3.17 ± 0.26 ^b^	4.40 ± 0.15 ^b^	1.75 ± 0.20 ^b^	5.97 ± 0.41 ^b^	8.85 ± 0.02 ^b^	1.38 ± 0.02 ^b^	2.91 ± 0.12 ^a^	63.82 ± 1.44	
CK-12	4.85 ± 0.05 ^a^	6.95 ± 0.29 ^a^	4.52 ± 0.90 ^a^	6.74 ± 0.22 ^b^	12.17 ± 0.71 ^a^	0.50 ± 0.03 ^a^	3.72 ± 0.67 ^a^	86.03 ± 4.67	
All-0	3.02 ± 0.05 ^c^	2.20 ± 0.18 ^c^	2.96 ± 0.11 ^a^	6.37 ± 0.24 ^a^	3.84 ± 0.04 ^c^	1.17 ± 0.01 ^b^	2.85 ± 0.10 ^b^	46.73 ± 1.80	
All-6	3.63 ± 0.09 ^b^	4.33 ± 0.07 ^b^	3.02 ± 1.54 ^a^	6.93 ± 0.23 ^a^	9.75 ± 0.07 ^b^	1.41 ± 0.02 ^a^	20.04 ± 0.00 ^c^	60.22 ± 3.16	
All-12	4.19 ± 0.08 ^a^	4.86 ± 0.03 ^a^	4.45 ± 0.06 ^a^	6.96 ± 0.09 ^a^	18.51 ± 0.05 ^a^	0.42 ± 0.00 ^c^	4.12 ± 0.16 ^a^	81.38 ± 1.04	
AOB-0	3.11 ± 0.02 ^a^	3.33 ± 0.02 ^b^	2.72 ± 0.02 ^c^	7.19 ± 0.01 ^a^	14.15 ± 0.08 ^b^	1.50 ± 0.03 ^b^	2.39 ± 0.76 ^a^	51.60 ± 1.53	
AOB-6	3.57 ± 0.13 ^a^	2.84 ± 0.19 ^b^	6.32 ± 0.14 ^a^	6.89 ± 0.24 ^ab^	5.57 ± 0.21 ^c^	1.73 ± 0.04 ^a^	2.47 ± 0.01 ^a^	47.87 ± 1.68	
AOB-12	4.33 ± 0.71 ^a^	4.49 ± 0.39 ^a^	3.76 ± 0.16 ^b^	6.63 ± 0.00 ^b^	17.86 ± 0.01 ^a^	0.33 ± 0.01 ^c^	3.09 ± 0.05 ^a^	72.56 ± 3.35	
AA-0	2.81 ± 0.01 ^c^	3.30 ± 0.00 ^b^	2.27 ± 0.01 ^c^	9.12 ± 0.10 ^a^	7.36 ± 0.13 ^a^	2.20 ± 0.03 ^a^	2.05 ± 0.24 ^b^	48.20 ± 2.66	
AA-6	3.06 ± 0.02 ^b^	3.43 ± 0.02 ^b^	2.56 ± 0.02 ^b^	8.02 ± 0.47 ^a^	5.67 ± 0.92 ^a^	20.03 ± 0.00 ^b^	2.66 ± 0.05 ^a^	50.48 ± 2.20	
AA-12	3.41 ± 0.06 ^a^	4.04 ± 0.11 ^a^	3.35 ± 0.12 ^a^	7.88 ± 0.51 ^a^	5.89 ± 0.03 ^a^	0.71 ± 0.00 ^c^	2.73 ± 0.12 ^a^	53.87 ± 2.10	

Different lowercase letters in superscript represent significant differences within groups (*p* < 0.05).

**Table 2 foods-12-03467-t002:** The thresholds of different amino acid tastes.

FAA	Taste	Threshold (mg/100 mL)
Asp	Umami	100
Glu	Umami	30
Ser	Sweet	150
His	Bitter	20
Gly	Umami/Sweet	130
Thr	Sweet	260
Arg	Sweet/Bitter	50
Ala	Umami/Sweet	60
Val	Sweet/Bitter	40
Met	Sweet/Bitter	30
Phe	Bitter	90
Ile	Bitter	90
Leu	Bitter	190
Lys	Sweet/Bitter	50
Pro	Sweet/Bitter	300

## Data Availability

The data used to support the findings of this study can be made available by the corresponding author upon request.

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
