# Peer review of "Synergistic Effect of Combined Treatment with Allicin and Antioxidant of Bamboo Leaves and Preservation of Bullfrogs (Lithobates catesbeiana) during Refrigeration Storage"

_foods, 2023, doi:10.3390/foods12183467_

Round 1

Reviewer 1 Report (Previous Reviewer 1)

To authors,

The revised version of the manuscript is fine and it provides enough experimental design for believable results. However, after re-checking the author's response on the similarity index test, this point was fine and had no influence on the data of the current manuscript. This version is ok for accepting to publication.

Best regards,

Reviewer 2 Report (Previous Reviewer 2)

The manuscript has been performed taking into account the previous comments and criticisms. I would recommend its acceptation.

Some minor performances could be done.

This manuscript is a resubmission of an earlier submission. The following is a list of the peer review reports and author responses from that submission.

Round 1

Reviewer 1 Report

The manuscript was found in a high level of plagiarism which was serious for the publication. So, this manuscript was rejected for a significant reason.

None

Reviewer 2 Report

The manuscript provides an interesting study including a combined treatment focused on the microbial and chemical quality performance of bullfrog leg quality.

I think some performances and clarifications ought to be carried out before a subsequent revision is done.

Abstract

Indicate the storage temperature.

Perform: “The reduction of TBA and TVB-N … “. It is their content that is reduced. This performance ought to be carried out throughout the whole manuscript.

What is the Control group ?

Keywords

Include: refrigerated storage and antimicrobial activity.

Material and methods

The concentrations chosen for allicin and bamboo leave antioxidant treatments ought to be justified.

What part of the body frog was studied ? The legs ? This ought to be expressed and mentioned throughout the paper.

No indication is provided on how the bamboo antioxidants were extracted. Also about the preparation of both solutions.

Group 4: Two subsequent treatments ? Or a single treatment including both solutions ?

The authors have provided very scarce information on their procedure. They have to consider that the reader may want to repeat the study.

Explain the procedure for TBARS determination, and how the quantification was carried out.

It is mentioned that measurements were made in replicate. This would give the error of the analytical procedure. But, were there replicates carried out concerning the technological process ? This is my main concern with this manuscript. How were the average values and standard deviations calculated ?

Results and discussion

Fig. 2, c: Y axis: TBARS content. Legend: Changes in TBARS content

Conclusions

… the augment of TBA, TVB-N … ? Please, perform.

Moderate performance ought to be carried out.

Reviewer 3 Report

  • The manuscript titled "Synergistic Effect of Combined Treatment with Allicin and Antioxidants of Bamboo Leaves on the Preservation of Bullfrog (Lithobates catesbeiana) During Refrigeration Storage" presents an interesting topic, but it requires significant improvements to meet the standards of scientific rigour and clarity. The following points highlight key areas that need attention:

    • Line Numbers: The manuscript lacks line numbers, which are essential for accurate referencing and ease of communication during the review process.

    • Terminology: The term "Polythilene" should be corrected to "Polyethylene" for accuracy.

    • References: Adhere to the journal's reference style guidelines. Ensure that author names are presented correctly, including capitalization, and remove any unnecessary brackets.

    • Equation Numbering: Equations should be numbered for easy reference throughout the manuscript.

    • Equation 1: The significance of "1-" in Equation 1 needs clarification. Provide an explanation or context for this element.

    • Amino Acid Determination: Specify the reference method used for amino acid determination to ensure the reproducibility of the study.

    • Materials and Methods: Strengthen the rigour of the Materials and Methods section. Provide reference methods or detailed explanations of calculations to enhance the transparency of the research process.

    • TPA Parameters: Specify the parameters measured in the Texture Profile Analysis (TPA) for better clarity.

    • Sensory Evaluation: Describe the training process of the sensory panellists and provide details about the reference and descriptors used in the sensory evaluation.

    • Probability Notation: Use lowercase "p" for probability, as per standard notation.

    • Results Presentation: Sections 3.2.1 and 3.4.1 should start with a description of the results rather than beginning with a figure.

    • Figure 2: The legend of Figure 2 mentions that letters have shifted in the image. Align the letters properly for accurate representation.

    • pH Fluctuations: Provide an explanation or reference for the statement regarding pH fluctuations caused by CO2 dissolution into the meat aqueous phase to form carbonic acid.

    • Figure 3: Consider splitting Figure 3 into a Table and a separate Figure, as it includes both a table and graphical components.

    • Thresholds: Clearly state the source or basis for the thresholds used as references.

    • Figure 6: Include a legend in Figure 6 for proper interpretation.

    • Section 3.7: Avoid starting Section 3.7 with a figure. Begin the section with a description of the results.

    • Figure 7: Clarify the purpose of Figure 7 and explain its inclusion in the manuscript.

    In summary, while the paper explores an intriguing subject, significant improvements are required in terms of clarity, scientific rigour, and proper referencing. 

The manuscript demonstrates a reasonable standard of English language, but it requires moderate editing to enhance its clarity and readability. While the overall writing is coherent, there are areas where sentence structures could be improved for smoother flow. Additionally, attention to grammar and punctuation will help eliminate minor errors and inconsistencies. By applying moderate editing, the manuscript's language quality can be notably improved, leading to a more polished and professional presentation of the research findings.